# Antioxidant Supplementation in Oxidative Stress-Related Diseases: What Have We Learned from Studies on Alpha-Tocopherol?

**DOI:** 10.3390/antiox11122322

**Published:** 2022-11-24

**Authors:** Fleur L. Meulmeester, Jiao Luo, Leon G. Martens, Kevin Mills, Diana van Heemst, Raymond Noordam

**Affiliations:** 1Department of Internal Medicine, Section of Gerontology and Geriatrics, Leiden University Medical Center, P.O. Box 9600, 2300 RC Leiden, The Netherlands; 2Department of Clinical Epidemiology, Leiden University Medical Center, P.O. Box 9600, 2300 RC Leiden, The Netherlands; 3NIHR Great Ormond Street Biomedical Research Centre, Great Ormond Street Hospital, UCL Great Ormond Street Institute of Child Health, 30 Guilford Street, London WC1N 1EH, UK

**Keywords:** antioxidant supplementation, α-tocopherol, oxidative stress, reactive oxygen species

## Abstract

Oxidative stress has been proposed as a key contributor to lifestyle- and age-related diseases. Because free radicals play an important role in various processes such as immune responses and cellular signaling, the body possesses an arsenal of different enzymatic and non-enzymatic antioxidant defense mechanisms. Oxidative stress is, among others, the result of an imbalance between the production of various reactive oxygen species (ROS) and antioxidant defense mechanisms including vitamin E (α-tocopherol) as a non-enzymatic antioxidant. Dietary vitamins, such as vitamin C and E, can also be taken in as supplements. It has been postulated that increasing antioxidant levels through supplementation may delay and/or ameliorate outcomes of lifestyle- and age-related diseases that have been linked to oxidative stress. Although supported by many animal experiments and observational studies, randomized clinical trials in humans have failed to demonstrate any clinical benefit from antioxidant supplementation. Nevertheless, possible explanations for this discrepancy remain underreported. This review aims to provide an overview of recent developments and novel research techniques used to clarify the existing controversy on the benefits of antioxidant supplementation in health and disease, focusing on α-tocopherol as antioxidant. Based on the currently available literature, we propose that examining the difference between antioxidant activity and capacity, by considering the catabolism of antioxidants, will provide crucial knowledge on the preventative and therapeutical use of antioxidant supplementation in oxidative stress-related diseases.

## 1. Introduction

Nutrition and other lifestyle factors have been shown to have an important impact on the incidence and outcomes of most of the common non-communicable diseases that have been associated with aging, such as neurodegenerative and cardiovascular diseases, type 2 diabetes and cancer [1]. Aging is a biological process of progressive decline in physiological functions with advancing chronological age, leading to an increased vulnerability to disease and, subsequently, death [2]. The characteristic functional changes that precede these diseases, such as physical impairment and cognitive decline, are driven by multiple biomolecular mechanisms, including the accumulation of cellular damage and epigenetic alterations, which collectively result in altered functioning at the cellular, tissue and organism levels [3,4]. These characteristic mechanisms have collectively been described as the “hallmarks of ageing” [5] and might comprise effective targets for preventive and curative treatments of multiple age-related disease conditions. 

Age-related diseases, such as neurodegenerative and cardiovascular diseases, type 2 diabetes and cancer, are affected by the hallmarks of aging [2]. Besides well-known pharmacological therapies such as statins, management of body weight and physical exercise have been shown to be preventive (lifestyle) strategies [6,7]. However, effective regulation of the age-associated cellular damage described through the hallmarks has not been accomplished yet.

One of the processes contributing to age- and adverse lifestyle related disease is mitochondrial dysfunction, of which oxidative damage may be an important cause and consequence [8]. The process of oxidative phosphorylation in the mitochondria produces reactive oxygen species (ROS). ROS encompass a group of molecules, either free radical or non-radical species, derived from molecular oxygen (O_2_) formed during reduction-oxidation (redox) reactions or by electronic excitation [9]. Free radicals have an unpaired electron, making them less stable and thus more reactive with various organic substrates than non-radical species. Non-radical species can, however, easily lead to free radical reactions in living organisms in the presence of transitions metals such as iron or copper [10]. Sources of ROS include endogenous sources (e.g., mitochondria, peroxisomes and NADPH oxidases) and exogenous sources (e.g., ultraviolet light, pollutants and ionizing radiation). These ROS can cause damage to macromolecules and mitochondria when the balance between ROS compounds and antioxidant defense mechanisms is disrupted. In turn, mitochondrial dysfunction will promote further free radical and non-radical ROS generation [9,11], for example, via the decreased expression of crucial proteins for electron transport due to damaged mitochondrial DNA (mtDNA). Oxidative stress refers to an “imbalance between oxidants and antioxidants in favor of the oxidants, leading to a disruption of redox signaling and control and/or molecular damage” [11]. Importantly, redox signaling by ROS compounds is required for normal cellular functioning and host defense mechanisms. When ROS generation is deficient or excessive, this may lead to a broad range of phenotypic changes including altered gene expression, cellular senescence and inhibited growth [9].

To prevent cellular damage and maintain ROS homeostasis, a complex system of different antioxidants exists. For example, antioxidant enzymes are involved in the neutralization of ROS in the mitochondria, including superoxide dismutase (SOD), catalase (CAT) and glutathione peroxidase (GPX). Non-enzymatic antioxidants comprise dietary vitamins such as vitamin C and vitamin E (α-tocopherol), which intercept free radical chain reactions. Alteration in acting antioxidant levels could result in a disruption of ROS production and removal, leading to disruption of ROS signaling or in oxidative-stress induced damage. Antioxidants have therefore been hypothesized to play an important role in the development of multiple diseases. In line with this hypothesis, a promising antioxidant in observational studies is α-tocopherol [12]. However, although many prospective cohort studies have observed associations between higher α-tocopherol levels and a lower risk of overall and chronic disease mortality [13,14,15], randomized clinical trials comparing α-tocopherol supplementation with placebo have failed to demonstrate any beneficial clinical effect of higher α-tocopherol levels on the onset and development of disease, particularly cardiovascular diseases [16,17,18,19].

To date, it remains difficult to make causal inferences about oxidative stress and the use of antioxidant supplementation in nutrition, and the implications in human health and disease. In the present review, we focus on the paradox of the therapeutic role of (dietary) antioxidants in disease with regard to the rapidly evolving field of nutrition and medical sciences, integrating important recent studies that used novel research techniques such as Mendelian randomization. Accordingly, we first provide a brief overview of the chemical processes resulting in oxidative damage and the role of (anti)oxidants, focusing on the non-enzymatic antioxidant α-tocopherol. We then summarize the pertinent evidence on antioxidant supplementation in both the general and disease population. The final part of the review addresses the controversy between the circulating levels and capacity of antioxidants and discusses directions for future research.

## 2. Process of Oxidative Damage in Health and Disease

### 2.1. Generation of Reactive Oxygen Species (ROS) 

The presence of ROS was first recognized in biological systems several decades ago [20]. ROS do not relate to a single species; rather, the term covers a range of small, short-lived molecules containing unpaired electrons formed by the partial reduction of O_2_ [20]. Of ROS molecules, non-radical hydrogen peroxide (H_2_O_2_) and typical free radicals of hydroxyl radical (•OH) and superoxide anion radical (O_2_^•−^) have been well studied and are considered among the key players in cellular damage [19].

The major endogenous enzymatic sources of ROS are transmembrane NADPH oxidases (NOXs) and the mitochondrial electron transport chain (ETC), as well as several other intracellular pathways involving cytosolic and membranal enzymes (e.g., cytochrome P450 enzymes, superoxide dismutase and monoamine oxidase) [9,21,22]. It is worth noting that the oxidation of polyunsaturated fatty acids generates lipid hydroperoxides and related radicals, alkoxyl and peroxyl, which impact redox signaling [23]. In addition to these endogenous sources, ROS are also produced from cumulative exposure to environmental factors such as nutrients, drugs, toxicants and physical or psychological stressors, albeit these exposures are highly variable [9,24].

O_2_^•−^, a free radical ROS, either dismutases spontaneously or deliberately via catalyzation by superoxide dismutates to H_2_O_2_ and O_2_ [25,26]. Hence, O_2_^•−^ serves as a major source of H_2_O_2._ This two-electron (non-radical) H_2_O_2_ is produced mainly by NOXs along with superoxide dismutases, as well as the mitochondrial ETC and many other enzymes [9]. H_2_O_2_ is a strong oxidant, but only reacts with a few biological targets including CO2/bicarbonate, which leads to peroxymonocarbonate (HCO_4_^-^) [9]. In turn, the most reactive ROS—free radical •OH—is formed by reduction of H_2_O_2_ in metal-catalyzed Fenton chemistry involving free iron (Fe^2+^) [25,26]. •OH reacts directly with the nearest neighboring biomolecule at the site of its generation, making the location of Fe^2+^, a strong determinant of the site of •OH toxicity. In summary, a key process of •OH generation can be schematically described as O_2_ → O_2_^•−^ → H_2_O_2_ → •OH. O_2_^•−^ also reacts efficiently with other radicals including nitric oxide (NO), of which peroxynitrite (ONOO^−^) is formed. Peroxynitrite can, in turn, modify proteins by the oxidation or nitration of amino acids such as tyrosine, leading to altered physical and chemical properties [27]. 

### 2.2. Implications of the ROS Balance in Maintenance of Health and Disease

#### 2.2.1. Physiological Range of ROS: Normal Cellular Functioning

For decades, research has mainly focused on the damaging effects of ROS due to their close association to age-related diseases [26]. However, ROS are also important components at a low to modest range in many redox-dependent processes to maintain cellular functions—considered as their “physiological range.” To define a certain physiological ROS range of such chemically diverse and transient species, both the beneficial and adverse effects of ROS should be taken into consideration. Nevertheless, the high rate of ROS generation and neutralization forms a challenge in determining this range. The chemical reactivity of the various ROS molecules is vastly different, extending up to 11 orders of magnitude in their respective second-order rate constants with particular targets [9]. Moreover, the range of physiological ROS may vary substantially between humans depending on numerous factors, including sex, age, nutritional and health status [28,29], and this range may vary between different time points even within one homogenous group, making it difficult to measure ROS population-wide or compare a certain ROS range directly between individuals.

The beneficial effects of ROS on cellular function and homeostasis are achieved via several signaling pathways [30]. For example, ROS may affect the activation of the nuclear factor kappa B (NF-κB) pathway via the inhibition of IκB𝛼 phosphorylation, inactivation of IκB kinase (IKK) or upstream kinases and interruption of the ubiquitination and degradation of I𝜅B by inactivating Ubc12 [31,32,33]. The mitogen-activated protein kinase (MAPK) cascade is also influenced by ROS compounds. Here, different ROS may activate members of the MAPK family by influencing their receptor or abolishing their inhibition, leading to intracellular signaling transduction essential to cell proliferation, differentiation, development, cell survival and apoptosis [30,34]. There are also strong links between oxidants and p53, which regulates cell cycle progression in response to a variety of stressors [35]. ROS may be implicated in the regulation and responsiveness of stress sensors by enhancing the antioxidant defense via p53 to maintain cellular redox balance and by indirectly modulating selective transactivation of p53 target genes [35].

Given the signaling effects of various ROS molecules, they play a pivotal role as second messengers in the maintenance of many cellular processes. Therefore, a small (albeit transient) increase in ROS levels within the physiological range may optimize cellular signaling and function, and thereby be beneficial for health [36]. As defense systems cannot eliminate all ROS before they react with macromolecules due to their extremely high reactiveness to specific molecules, even in a healthy situation [37], some oxidative damage to cells is always produced. The constantly changing dynamics in ROS exposure of the human body can be best described via an optimal curve, as discussed in detail previously [22]. When ROS production and removal remain within the physiological range, this will have beneficial effects on the functioning of the human body. However, when the body cannot adapt to the decrease or increase in ROS leading to dysregulated ROS homeostasis, this will result in adverse effects (Figure 1).

#### 2.2.2. Pushing the Boundaries of the ROS Balance

ROS beyond the physiological range may irreversibly react with macromolecules, including DNA, proteins and lipids, causing them to lose their function or gain inappropriate functionalities [20]. In turn, these damaged macromolecules may accumulate intracellularly and accelerate age-related diseases. 

As is clear from previous research, deficient or excessive levels of ROS molecules are associated with a wide range of diseases [37,38,39,40]. However, an unidentified grey area remains in which various ROS may contribute to accelerated aging and diseases without exhibiting a clear disease phenotype. Individuals with a small (subclinical) increase in ROS over a longer period of time, either by overproduction, inadequate counter mechanisms or a combination of the two, may experience a constant level of moderate oxidative stress level on their tissues. In line with this hypothesis, oxidative stress has been suggested to play a role in multiple diseases [37,41], including cardiovascular disease, neurodegenerative disease and cancer.

#### 2.2.3. The Role of Excessive ROS in Cardiovascular Diseases

The vascular endothelium is crucial in preserving vascular function, making endothelial dysfunction a major initial cause of cardiovascular disease (CVD). Considering that endothelial function is partly regulated by redox components, excessive ROS have been associated with increased risks of cardiac hypertrophy, heart failure and atherosclerosis [41,42,43,44,45]. From a physiological point of view, cardiac myocytes are more susceptible to high ROS production than other, less energy-demanding, cells due to their relatively high number of mitochondria [46]. Although physiological low levels of H_2_O_2_ by NOX4 are required for vasodilation, normal endothelial function and vascular remodeling, supraphysiological H_2_O_2_ levels have the opposite, adverse effect: vasoconstriction, endothelial dysfunction, hypertension and increased inflammation [9].

Lipid peroxidation, a process involved in oxidative stress, contributes to the development of atherosclerosis and other CVDs. For example, malondialdehyde, a lipid peroxidation-derived aldehyde, can induce proinflammatory responses and contribute to the activation of the complement system in atherosclerosis [47]. Furthermore, 4-hydroxynonenal (4-HNE), generated by the decomposition of arachidonic acid and larger polyunsaturated fatty acids, has been implicated in the regulation of autophagy during myocardial ischemia and reperfusion. Accordingly, suppression of 4-HNE-stimulated autophagy in mice transfected with aldehyde dehydrogenase 2, a major enzyme involved in neutralization of 4-HNE, has been reported to reduce myocardial dysfunction [48]. In addition, ROS molecules contribute to endothelial damage and the consequential transformation of recruited macrophages into atherosclerotic plaque-forming foam cells by promoting the oxidation of low-density lipoproteins (LDL) [49]. ROS also induce the release of matrix metalloproteinases (MMPs), which promotes physical disruption of the atherosclerotic plaque and thereby exacerbates atherosclerosis development [50].

Similar to other unsaturated lipids, cholesterol is also susceptible to oxidative modification [51]. These oxygenated derivatives of cholesterol (oxysterols) present a remarkably diverse profile of biological activities, including apoptosis and platelet aggregation. The accumulation of oxysterols has been implicated in oxidative stress-related pathophysiology. For example, oxysterols are found enriched in pathologic structures such as macrophage foam cells and atherosclerotic lesions [52]. Notably, oxysterols have been shown to enhance MMP-9 levels and activity in human cells of the macrophage lineage through the induction of NOX2 activity, hence contributing to atherosclerotic plaque erosion and rupture, as well as ROS production [53]. However, despite their harmful proinflammatory features, oxysterols are currently emerging as fine regulators of physiological processes, including those involved in aging [54]. For example, at submicromolar concentrations, oxysterols have been reported to have anti-inflammatory activity. Oxysterols may also regulate cell death and protein homeostasis. Nevertheless, the impact of oxysterols on biological processes under physiological circumstances remains to be explored in more detail.

Mitochondrial dysfunction has also been directly linked to CVD. For example, a lower mtDNA copy number in lymphocytes, as a rough proxy of mitochondrial dysfunction, has been associated with higher CVD risk in large prospective studies, and the association between low mtDNA copy number and coronary artery disease is likely to be causal [50].

#### 2.2.4. The Role of Excessive ROS in Neurodegenerative Diseases

Neurodegenerative diseases (NDDs) are characterized by the progressive loss of neurons [55]. Neuronal cells are particularly vulnerable to oxidative stress due to a combination of high energy and oxygen demand, low antioxidant activity, a high number of cells in the post-mitotic state, abundant lipid content and a limited capacity of cell renewal [56]. Misfolded proteins aggregate and accumulate in the brain and contribute to neurodegeneration [55], for example, via the upregulation of NOX activity and oxidant generation [9]. In fact, several of these proteins are connected to mitochondrial (dys)function and associated with the production of ROS compounds. For example, Alzheimer’s disease (AD) may originate from deregulation of the redox balance [57]. In AD, lipid peroxidation, where lipids (e.g., in the myelin sheets) are oxidized by ROS, is greatly enhanced, especially in the amygdala and hippocampus [57]. The products of lipid peroxidation often cause crosslinked molecules (e.g., collagen) that are able to resist intracellular degradation and cause altered cellular communication. In addition, increased levels of sporadic (unique) mutations have been found in the mtDNA of AD patients [57,58]. Of specific interest, several of these mutations cause decreased transcription levels of essential mitochondrial proteins in AD. In the case of Parkinson’s disease (PD), studies have demonstrated a reduced activity of mitochondrial complex I in the dopaminergic neurons of the substantia nigra of PD patients, presumably contributing to excessive ROS generation accounting for the apoptosis observed in this part of the brain [55,59].

#### 2.2.5. The Role of ROS in Cancer Pathogenesis

Oxidant generation is strongly linked to initiation, progression and bystander effects in the tumor microenvironment, as well as to the biology of metastasis [21,60]. The role of ROS in cancer pathogenesis appears to be dependent on the stage of the tumor. In the early stages of cancer, ROS have been considered to have a pro-oncogenic role. As previously mentioned, ROS may modulate the selective transactivation of target genes of the tumor suppressor p53 [35]. Moreover, loss-of-function mutations in p53 may induce a further increase in intracellular ROS, provoke abnormal mitosis and promote cancer development. The increased production of ROS by cancer cells was shown to eventually support proliferation and allow cancer cells to adapt to stress due to a lack of nutrition or hypoxic environment [9,61,62,63].

On the other hand, ROS may exhibit a tumor-suppressor role during the later stages of cancer. It was shown that the expression and activity of antioxidant enzymes were increased in malignant tumors compared to adjacent normal tissue [64]. However, this enhanced activity of antioxidant systems in tumors has been associated with chemotherapy resistance [62,65]. Considering that the antioxidant activity increases in later cancer stages, the excessive intratumor oxidative damage is limited, which, in the end, aids the cancer cells to escape apoptosis. Accordingly, studies have investigated the effect of ROS-scavenging antioxidant supplementation, such as high-dose (pharmacological) ascorbate, on cancer development [66,67]. Nevertheless, the results of these studies on the benefits and adverse effects of antioxidant supplementation in tumor progression remain inconsistent and require further investigation.

## 3. Role of Antioxidants in ROS Elimination

### 3.1. Working Mechanisms of Antioxidants

A complex defense mechanism to compensate for ROS generation consists, among other mechanisms, of multiple antioxidants. Antioxidants are compounds that inhibit oxidation, thereby delaying or inhibiting cellular damage [68]. The main antioxidants are either formed endogenously (glutathione, reduced coenzyme Q, uric acid, bilirubin) or are diet-derived, for example, from plant oils, nuts, and seeds (α-tocopherol (vitamin E)), (citrus) fruits and vegetables (ascorbate (vitamin C), carotenoids) [68,69]. Although it should be noted that antioxidants may not outcompete the dedicated enzymes that can catalytically deplete ROS (e.g., SOD, CAT and GPX), the mechanisms of antioxidants, such as α-tocopherol, have been researched [69,70]. Antioxidants may also be classified based on their activity, which includes enzymatic or non-enzymatic antioxidant activity. Enzymatic antioxidants catalyze the conversion of oxidized metabolic products to stable, nontoxic molecules, whereas non-enzymatic antioxidants intercept free radical chain reactions [71]. Although the individual roles of antioxidants in the human defense system are divergent, antioxidants act in a cooperative and synergistic manner, involving a complex network of interacting compounds [68,69]. 

The protecting actions of antioxidants can be described as two principal mechanisms that act simultaneously [68,69]. First, antioxidants prevent the formation of ROS via quenching oxygen molecules or sequestering active metal ions, including iron (Fe; II/III) and copper (Cu; I/II). In addition, antioxidant enzymes work to catalytically deplete ROS. For example, SOD catalyzes the dismutation of two molecules of O_2_^•−^ to H_2_O_2_ and molecular oxygen, and GPX prevents the harmful accumulation of H_2_O_2_ by catalyzing the conversion of H_2_O_2_ to water [71,72]. The activities of these antioxidant enzymes may, however, change during aging. For example, an age-related reduction in SOD and CAT gene expression was observed in the granulosa cells from periovulatory follicles in women [73], and a progressive decrease in SOD, CAT and GPX activity was observed in erythrocytes of older individuals when compared to younger individuals (55–59 y/o) [74]. 

The second protective mechanism of antioxidants concerns the chain-breaking antioxidants. These compounds contribute to the elimination of ROS compounds before they may irreversibly react with and impair biological macromolecules, for example, in lipid peroxidation. Chain-breaking antioxidants can either receive an electron from a radical or donate one in order to terminate the chain reaction, resulting in the formation of stable by-products [68,69]. When these two protective mechanisms appear insufficient to prevent oxidative damage by ROS, antioxidants and enzymes can repair the resultant damage and reconstitute the harmed tissues. The repair systems’ intervention includes restoring oxidatively damaged nucleic acids, removing oxidized proteins via intra- and extracellular proteasomal systems and repairing oxidized lipids. Together, antioxidants provide a complex safety net to cope with the constant generation of various ROS molecules. As α-tocopherol is one of the most well-studied antioxidants, this review mainly focuses on the role of α-tocopherol in health and disease.

### 3.2. Antioxidant Supplementation in Age-Related Diseases

Hypothetically, increasing antioxidant levels in individuals with excessive ROS should alleviate the associated development of diseases by supporting the restoration of the ROS balance within the optimal physiological range. One way to effectively enhance functioning antioxidant levels is via dietary supplementation. Most epidemiological cohort studies have found protective effects of increased dietary or circulating levels of antioxidants on lower disease incidences [75]. For instance, several epidemiological cohort studies have shown that higher intake of antioxidants, either via regular diet or as oral (over the counter) supplements, were associated with a lower risk of incident CVD [76,77]. In addition to CVD, higher intake of antioxidants or supplements has been associated with a lower risk of incident Alzheimer’s disease [78,79], Parkinson’s disease [80] and amyotrophic lateral sclerosis (ALS) [81,82] in a number of prospective cohort studies.

The results from prospective cohort studies led to the concept of antioxidant supplementation in the general population, as it may ameliorate or even prevent several age-related diseases. However, evidence from clinical trials supporting the clinical benefit of the use of antioxidant supplements in the general population is still lacking. An example is the Women’s Health Study, in which approximately 40,000 healthy US women aged 45 and older were randomly assigned to receive α-tocopherol or placebo and subsequently followed the treatment for more than 10 years [18]. Based on the results, the authors concluded that daily intake of α-tocopherol did not provide the overall clinical benefit for major CVD events or cancer. Moreover, the group taking α-tocopherol supplements did not show a lower risk of (cardiovascular) mortality. A similar result was seen in the Physicians’ Health Study II and HOPE study, which examined a combination α-tocopherol and vitamin C supplementation; no reduced risks of major incident cardiovascular events [17] or cancer [83] were observed.

In addition to the conventional study designs, we previously implemented a Mendelian randomization (MR) framework to investigate the relationship between dietary-derived circulating antioxidants and CVD [84,85]. In MR studies, genetic variants are used as instrumental variables to infer causality of lifelong exposure to certain risk factors on diseases (outcome), as illustrated in Figure 2. As the genetic information is fixed at conception, MR is not affected by most confounding factors and reverse causation, which are the main limitations from prospective cohort studies. In our recent work comprising over 700,000 participants with more than 93,000 coronary heart disease cases, genetically predicted circulating dietary-derived antioxidants were unlikely to be causal determinants of primary CHD risk [84]. Similarly, in over one million individuals, no evidence was found for a causal association between dietary-derived circulation antioxidants and ischemic stroke [85]. In the context of neurodegenerative diseases, similar null findings were obtained between vitamin A, vitamin C, β-carotene, and urate and risk of AD [86]. Taken together, these genetic studies do not support the beneficial role of dietary-derived antioxidants on disease risks in the general population.

### 3.3. Antioxidant Supplementation in α-Tocopherol-Deficiency

The discrepancy in study results between the prospective studies on one hand and the randomized clinical trials and MR studies on the other hand may be related to differences in the study populations. Notably, the beneficial effects of antioxidants were mostly demonstrated in patients with extreme local concentrations of ROS or a deficiency in their antioxidant production and/or metabolism [79,82,87]. An example of an antioxidant deficiency disease in humans where supplements may provide health benefits is ataxia with isolated vitamin E deficiency (AVED), a rare inherited neurodegenerative disorder that affects approximately fewer than one in one million individuals [87,88]. AVED is induced by mutations in the gene coding for α-tocopherol transfer protein (α-TTP), which is required for α-tocopherol retention [69,89]. α-tocopherol deficiency can develop secondary disorders that cause an impaired absorption of α-tocopherol from adipose tissue. AVED is characterized by low plasma α-tocopherol levels, which can be increased through α-tocopherol supplements to normal levels [87]. Accordingly, a study investigating the effect of α-tocopherol supplementation on AVED disease status observed reduced disease progression after a 12-month treatment [88]. 

Apart from the observed results on AVED disease status, it is worth mentioning that beyond the antioxidant effect of α-tocopherol, there may be biological effects unrelated to its chain-breaking antioxidant actions that may contribute to the AVED phenotype. Considering that ascorbic acid deficiency causes the clinical syndrome scurvy due to its role in collagen synthesis [90], which is treated with supplemental vitamin C, the beneficial effects of supplementing α-tocopherol in α-tocopherol-deficient individuals may not be solely due to its antioxidant effects. For example, α-tocopherol was shown to inhibit protein kinase C (PKC) and has been associated in a number of cellular events that are related to non-antioxidant properties of α-tocopherol, including cell proliferation, cell adhesion, enhancement of the immune response and gene expression [70].

### 3.4. Antioxidant Supplementation in the General, Healthy Population

Most of the aforementioned randomized clinical trials investigating antioxidants included healthy individuals from the general population. Importantly, although healthy individuals in the general population may also occasionally experience lower antioxidant levels, these levels may overall still be sufficient to cope with the constant production of oxidants, causing the antioxidant supplements to have no effect on lowering disease risk. Generally, only very few individuals included in these studies had excessively high antioxidant levels. For this reason, supplementing antioxidants might not induce a sufficient clinical effect that can be detected in the statistical analyses. 

This hypothesis is supported by our recent work performed in the Netherlands Epidemiology of Obesity (NEO) study [91]. This population-based, prospective cohort study included individuals between 45 and 65 years of age living in the greater area of Leiden, the Netherlands (*N* = 6671). We were particularly interested in the associations between observed levels of α-tocopherol in serum and its metabolomics in urine in relation to behavioral and (subclinical) disease outcomes in a random subsample of 520 individuals. In several studies, the associations between α-tocopherol serum levels and lifestyle factors (such as smoking and alcohol use [92]), measures of glucose homeostasis [93], measures of body fat [94] and lipoprotein (sub)particles [95] were investigated. Overall, these studies found no associations, or even trends, between circulating α-tocopherol in serum and the different study outcomes. This could be due to the relatively small study population included from the NEO study. However, since only a few cases of obesity-related disease or mortality had been documented over the course of a 10-year follow-up, it is plausible that the included participants were relatively healthy. This supports the hypothesis that increasing α-tocopherol levels, particularly via the intake of supplements, does not have an effect on the health status of the general population. As long as an individual’s α-tocopherol level at baseline can adequately lower ROS generation and eliminate produced ROS, exceeding baseline levels with supplements may have little clinical effect.

In addition, the associations between different α-tocopherol urinary metabolites and serum α-tocopherol and lifestyle factors have been investigated previously. Since the metabolism of α-tocopherol can follow two pathways, it forms either the oxidized metabolite α-tocopheronolactone hydroquinone (α-TLHQ) or enzymatic metabolite of α-carboxymethyl-hydroxychroman (α-CEHC) that is measured in urine [94]. A-TLHQ is the oxidized metabolite generated when lipid peroxidation is successfully inhibited by α-tocopherol, representing ROS scavenging-dependent reactions, whereas α-CEHC is the product of enzymatic conversion of α-tocopherol in the liver. These metabolites were measured as sulfated and glucuronidated conjugates of α-tocopherol, the main forms of vitamin E, by mass spectrometry analyses of NEO urine samples. In the NEO study, circulating α-tocopherol did not correlate with its oxidized but with the enzymatic metabolite in urine [93]. This may suggest that the circulating α-tocopherol level was not a rate-limiting step for the conversion to its oxidized metabolites. Therefore, α-TLHQ is depicted as a marker of oxidative stress, while α-CEHC represents α-tocopherol status [94]. It was hypothesized that higher levels of α-TLHQ would be associated with higher disease risk and adverse lifestyle. Indeed, an association between current smokers and higher α-TLHQ levels compared with non-smokers was observed [92]. However, these studies also showed some contradictory results that urinary oxidized α-tocopherol metabolites were moderately associated with reduced insulin resistance [93] and marginally associated with lower body mass index, total body fat and visceral adipose tissue [94]. These findings provide remarkable insights on the role of α-tocopherol in health and disease and may suggest the urinary metabolite levels could instead reflect antioxidant capacity (e.g., lower levels of urinary metabolites as a marker of lower oxidant scavenger capacity).

### 3.5. Antioxidant Circulating Levels Versus Antioxidative Capacity

Given that serum α-tocopherol and its metabolites were not correlated in the previously described studies [93], observed the circulating levels of α-tocopherol—particularly induced by its synthetic forms, which have a lower bioavailability than natural α-tocopherol [96]—may not reflect the actual α-tocopherol activity. It should be emphasized that although the terms “antioxidant activity” and “antioxidant capacity” are often used interchangeably, they have different implications [68]. Notably, the antioxidant bioactivity of circulating levels refers to antioxidant kinetics in which a characteristic of a specific antioxidant is expressed as a value of the reaction rate times the reaction volume. The antioxidant capacity is rather defined as the measure of the total amount of oxidants scavenged via antioxidant mechanisms, which indicates the sum of antioxidant activity of the human body [97]. The bioactivity of α-tocopherol and other antioxidants can be influenced by several factors, including the intake of competing nutritional factors, absorption and metabolism, as well as genetics, age and lifestyle [98]. Therefore, it is plausible that only measuring (unmetabolized) antioxidants in blood is not sufficient to make inferences about antioxidant status. This hypothesis could explain why targeting antioxidant capacity by solely increasing circulating antioxidant levels, for example, via oral supplements, does not yield any clinically significant reductions in disease risks. Targeting the metabolism of antioxidants to oxidized or enzymatically converted metabolites may provide essential knowledge on antioxidant working mechanisms in the body, which may serve as a marker in future trials to monitor antioxidant utility after supplementation. This hypothesis should be examined in greater detail, preferably in larger study samples.

## 4. Discussion and Concluding Remarks

### 4.1. Antioxidant Supplements: Is There Really Any Benefit?

To date, there is an ongoing controversy about the use of antioxidant supplements for the prevention and treatment of multiple diseases. There is ample molecular evidence: an imbalance in ROS production and elimination can lead to oxidative damage, which triggers a cascade of the hallmarks of ageing and may contribute to the onset and development of numerous diseases [5,20,37,99]. Rationally, research has subsequently focused on enhancing the system that can effectively eliminate ROS: the complex network of antioxidants. Although it may seem only reasonable that increasing antioxidant levels to eradicate excessive ROS molecules should alleviate the burden caused by the overproduction of various ROS compounds, randomized clinical trials and MR studies to date have failed to provide evidence supporting this rationale [17,18,83,84,85,86]. A large discrepancy exists between the molecular indication and clinical outcomes for antioxidant supplementation. Therefore, the question is whether antioxidant supplementation truly provides considerable benefits on health status. Notably, the intake of antioxidant supplements as a therapy for low antioxidant status, due to, e.g., antioxidant deficiency diseases, may improve the patients’ health status and quality of life. However, this category of exceptionally low antioxidants levels only covers a small part of the dynamic and transient range of ROS. The greater part of the range of ROS, where defense mechanisms are sufficient for efficient ROS elimination, can be identified in the general population. These individuals with adequate antioxidant levels at baseline may only increase the circulating levels of antioxidants through the intake of supplements, but not the actual antioxidant capacity to eliminate part of the produced ROS. In other words, the network of antioxidant compounds may not become more effective by augmenting the pool of individual antioxidants with supplements in the general population (Figure 3).

The balance between ROS and antioxidants can also tilt toward excessive antioxidant levels (Figure 3, left panel). Through increased endogenous production, enhanced daily food intake or a combination of the two, antioxidant levels could theoretically exceed its healthy boundaries and cause adverse effects. Although little is known about the possible detrimental effects of antioxidant supplementation, non-enzymatic antioxidants, including vitamin C and α-tocopherol, have been shown to have pro-oxidant effects at high concentrations, leading to ROS generation and contributing to a state of oxidative stress [100,101]. It has also been shown that α-tocopherol may interact with other vitamins to enhance or interfere with their function [102]. Accordingly, α-tocopherol can interfere with the blood clotting capacity of vitamin K [102], resulting in reduced blood clot formation. Although this aspect may be beneficial in certain patients, including in women with recurrent abortion due to impaired uterine blood flow [103], it may also increase the risk of bleedings in healthy individuals. However, it is important to consider that these adverse clinical effects of α-tocopherol antioxidant use could also be observed due to chance or possible flaws in the study design and/or selection of the study population. Taken together, these results indicate that antioxidant supplementation, particularly α-tocopherol, should be used with caution for adverse effects. It is therefore important to determine whether an individual genuinely requires antioxidant supplementation before intake.

To this end, it is essential to measure oxidative damage markers and ROS turnover in the human body. However, measuring these endpoints forms a challenge in research. No single parameter has been recommended as a gold standard for measuring redox status in clinical studies thus far [104]. A major limitation is the identification of reliable biomarkers [105]. Some biomarkers have been identified in experimental and population-based epidemiological studies. Examples of current biomarkers for lipid peroxidation include plasma malondialdehyde, 4-hydroxynonenal and isoprostanes; for nucleic acid oxidation, examples include 8-oxo-7,8-dihydro-2’-deoxyguanosine (8-oxodG) and 8-oxo-7,8-dihydroguanosine (8-oxoG) for DNA and RNA, respectively [106]; and protein carbonyl can be used as a biomarker of protein oxidation [107]. Despite the potency of measuring these biomarkers of oxidative stress, the measured oxidative damage is often the result of a complex, interacting mechanism of numerous endogenous and exogenous antioxidants. Furthermore, these biomarkers cannot reflect the complete oxidative damage that has been brought to the body since they are mostly exclusive to certain macromolecule damage [22]. In addition to biomarkers, measuring ROS as a representation of oxidative stress has its limitations [104]. Some ROS molecules are highly reactive (particularly hydroxyl radicals) and therefore have a relatively short half-life, which makes their measurement in biological systems a complex task. Since accurate measurements of pro- and antioxidant levels are crucial to make inferences about the use of antioxidant supplementation, it is important to define an integrative yet clinically applicable approach to determine an individual’s redox status.

### 4.2. Final Remarks and Conclusions

Regarding the key role of oxidative damage in ageing and the onset and development of several diseases, research on decreasing oxidative damage with antioxidants has emerged in the last few decades. However, since clinical trials to date have not supported the use of antioxidant supplementation in oxidative stress-related diseases, a paradox exists: does supplementation of antioxidants delay aging and/or treat oxidative stress-related diseases? 

In summary, there are three critical points to consider when examining the use of antioxidant supplementation. First, identifying reliable biomarkers for antioxidant capacity and levels of oxidative species that reflect the overall redox status in vivo, as well as transient redox status in specific tissues or cells, is crucial for further research. To date, there is still little consensus about the gold standard for measurements of oxidative stress in vivo. An optimal biomarker should be easily accessible, simple to detect accurately in human tissue and/or body fluid and reasonably stable. Second, the difference between antioxidant activity and capacity should be recognized in further research. Supplementation of antioxidants may increase their circulating levels and bioactivity, but this does not imply that the capacity of antioxidants is enhanced. Furthermore, several mechanisms may contribute to the difference between antioxidant activity and capacity, including its metabolism. Third, regarding the physiological importance of ROS signaling, it is necessary to develop strategies in redox studies that selectively address disease-associated mechanisms without disrupting the signaling pathways of ROS compounds. Future research should therefore focus on exploring novel markers for measuring oxidative stress and antioxidant status in vivo. Reliable yet simple measurements can facilitate in-depth studies examining the effects of antioxidant supplementation in aging and the development and progression of oxidative stress-related diseases, as well as in the general population, providing crucial knowledge that is indispensable to make inferences about the use of antioxidant supplements by healthy and diseased individuals.

## Figures and Tables

**Figure 1 antioxidants-11-02322-f001:**
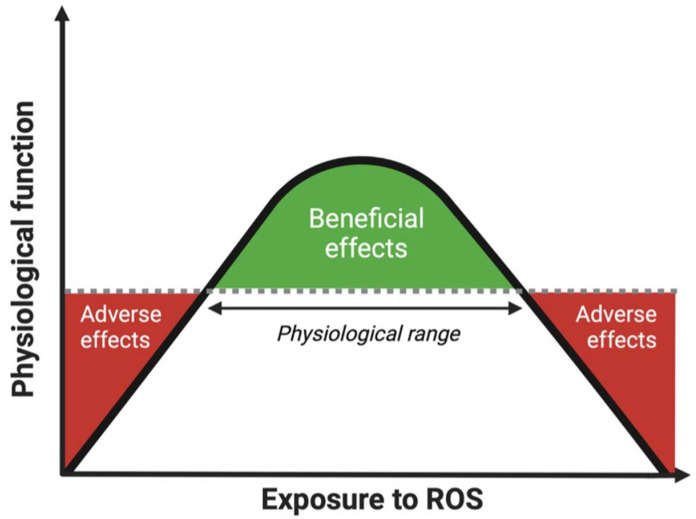
Optimal curve of the effect of exposure to ROS on physiological function of the human body. ROS within the physiological range have beneficial effects on physiological function. When the exposure to ROS in the human body goes beyond the physiological range, either too low or too high, this may lead to adverse effects and thus reduced physiological functioning. Abbreviations: ROS, reactive oxygen species.

**Figure 2 antioxidants-11-02322-f002:**
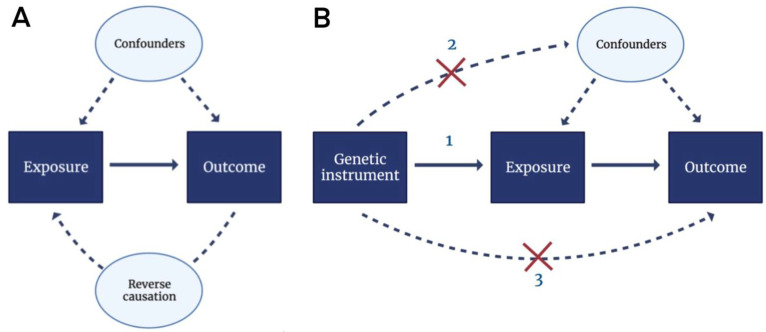
Directed acyclic graphs (DAGs) for observational studies and Mendelian randomization analyses. (**A**) In observational studies, the association between the exposure (e.g., antioxidant supplementation) and the outcome (e.g., incident CVD) was assessed. However, unmeasured confounding and reverse causation may have occurred in this set-up. Confounding factors influence both the exposure and the outcome, whereas reverse causation denotes an occasion where the outcome affects the exposure instead of vice versa. (**B**) Unmeasured confounding and reverse causation can be overcome by Mendelian randomization using genetic instruments (e.g., SNPs). Three assumptions apply: (1) genetic instruments are associated with the exposure, (2) genetic instruments are not associated with confounders and (3) genetic instruments affect the outcome only through the exposure. Abbreviations: CVD, cardiovascular disease; SNP, single nucleotide polymorphism.

**Figure 3 antioxidants-11-02322-f003:**
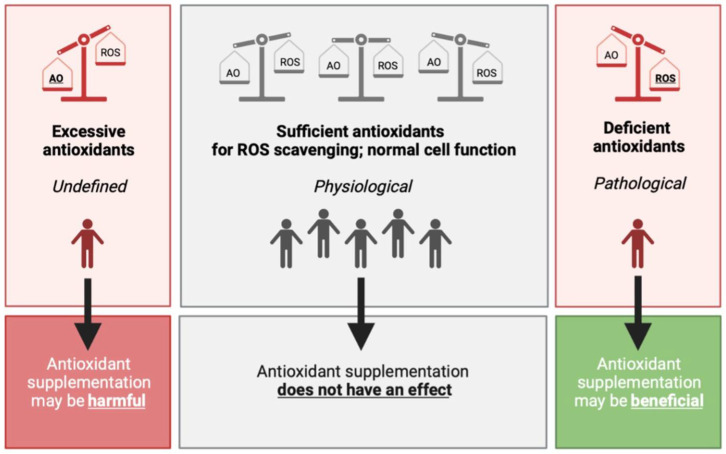
The hypothesized effect of antioxidant supplementation on the balance between antioxidants and ROS. (**Left**) In the case of excessive antioxidant compounds, either via endogenous production or via daily food intake, and/or antioxidant enzyme activity, lower oxidant levels can be observed with potentially harmful effects on normal cell function and communication. (**Middle**) Upon physiological levels, where antioxidants levels are sufficient to contribute to the removal of possibly harmful ROS, it is hypothesized that antioxidant supplementation does not have an effect on (oxidative stress-related) disease risk. This group comprises the greater part of the population, which are generally healthy individuals. (**Right**) Excessive ROS production may cause damage to intracellular macromolecules. When the damage is not able to be cleared by the repair system of the body, pathological changes or manifestation may occur. Antioxidant supplementation therefore may have beneficial clinical effects on aging and oxidative stress-related diseases. Abbreviations: AO, antioxidant; ROS, reactive oxygen species.

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
