# Peer review of "Antioxidant Supplementation in Oxidative Stress-Related Diseases: What Have We Learned from Studies on Alpha-Tocopherol?"

_antioxidants, 2022, doi:10.3390/antiox11122322_

Round 1

Reviewer 1 Report

The authors of "Antioxidant supplementation in oxidative stress-related diseases: what have we learned and what remains to be explored?" have written a review with a bit different perspective on theme titled. In my opinion - the title lacks one of key players of this review - α-tocopherol - since it is almost only one referred.

 There are few errors in chemical formulas in text that should be corrected appropriately (all marked in yellow in the original pdf file.)

The paper emphasizes three critical points when evaluating antioxidant supplementation: reliable biomarkers, difference between antioxidant capacity and activity and ROS signalling homeostasis. All of these points have merit in assessing the need for antioxidant supplementation - hence - I recommend this paper to be published.

Reviewer 2 Report

The review is interesting, but I think there are numerous issues that must be clarified and corrected.

Title: Antioxidant supplementation in oxidative stress-related dis-2 eases: what have we learned and what remains to be explored? The last part is not found in the review - what remains to be explored. Only the conclusions section contains a very vague sentence that defies this aspect of the title.   

Introduction:

-       Lines 54 – 65:  Give a brief description of the free radicals and non-radical species categories of ROS. Which of these is a result of dysfunctional mitochondria? At the molecular level, aren't there additional factors?

-       Lines 66 – 68: Additionally, the mitochondrial antioxidant enzymes should be defined in this section. The first line of defense against oxidative stress caused by dysfunctional mitochondria involves antioxidant enzymes.

Process of oxidative damage in health and disease:

-       Lines 108 – 109 – In addition to the significance of iron ions, it is necessary to specify which specific substrates (superoxide and hydrogen peroxide) are involved in the Fenton reaction.

-       Lines 189 – 202: The involvement of cholesterol oxidation products must be specified. One of the most significant peroxidation processes that occurs in plasma and LDL is the oxidation of cholesterol. Even though cholesterol oxidation is unquestionably a sign of increased oxidative stress under specific pathophysiological circumstances, this process appears to occur to some extent in healthy individuals. Oxysterols can diffuse through membranes considerably more quickly than cholesterol itself. Both free oxysterols and oxysterols esterified with long-chain fatty acids can be found in tissues. For instance, the esterified forms of oxysterols make for 80–95 % of all oxysterols in human atherosclerotic lesions. (for more data, see for example the article Oxysterols and Their Cellular Effectors, doi:10.3390/biom2010076).

-       Lines 228 – 239: The chapter "The role of ROS in cancer pathogenesis" is much too general and treated superficially. It should be improved – see for example doi:10.1152/ajpregu.00247.2017.

Role of antioxidants in ROS elimination:

-       Line 261: In my opinion the most important proteins responsible for sequestering active metal ions should be specified.

Discussion and concluding remarks:

-       Their pro-oxidant function under specific conditions must also be mentioned when discussing non-enzymatic antioxidants.

-       Lines 486 – 502: oxidative damage markers –Oxidative stress markers do not only refer to the products of lipid peroxidation. It is important to mention the markers utilized to identify certain nucleic acid oxidation products as well as protein oxidation products. They are equally important.

Round 2

Reviewer 2 Report

The resubmitted article's quality has improved, but there are still some aspects to be corrected.

The title of this publication has been changed by the authors to include this tocopherol aspect. However, tocopherol is not mentioned in the abstract or the introduction.
